# Mechanistic insight into the functional role of human sinoatrial node conduction pathways and pacemaker compartments heterogeneity: A computer model analysis

**Jichao Zhao**[1]\*, **Roshan Sharma**[1], **Anuradha Kalyanasundaram**[2], **James Kennelly**[1], **Jieyun Bai**[1], **Ning Li**[2], **Alexander Panfilov**[3], **Vadim V. Fedorov**[2]\*

1 Auckland Bioengineering Institute, University of Auckland, Auckland, New Zealand, 2 Department of Physiology & Cell Biology, Bob and Corrine Frick Center for Heart Failure and Arrhythmia; The Ohio State University Wexner Medical Center, Columbus, Ohio, United States of America, 3 Gent University, Gent, Belgium

\* j.zhao@auckland.ac.nz (JZ); vadim.fedorov@osumc.edu (VVF)

**Data Availability Statement:** The two original human SAN at DOI: 10.1113/JP273259 and RA at DOI: 10.3389/fphys.2017.00946 cellular models

## Abstract

The sinoatrial node (SAN), the primary pacemaker of the heart, is responsible for the initiation and robust regulation of sinus rhythm. 3D mapping studies of the ex-vivo human heart suggested that the robust regulation of sinus rhythm relies on specialized fibrotically-insulated pacemaker compartments (head, center and tail) with heterogeneous expressions of key ion channels and receptors. They also revealed up to five sinoatrial conduction pathways (SACPs), which electrically connect the SAN with neighboring right atrium (RA). To elucidate the role of these structural-molecular factors in the functional robustness of human SAN, we developed comprehensive biophysical computer models of the SAN based on 3D structural, functional and molecular mapping of ex-vivo human hearts. Our key finding is that the electrical insulation of the SAN except SACPs, the heterogeneous expression of $I_f$, $I_{Na}$ currents and adenosine A1 receptors (A1R) across SAN pacemaker-conduction compartments are required to experimentally reproduce observed SAN activation patterns and important phenomena such as shifts of the leading pacemaker and preferential SACP. In particular, we found that the insulating border between the SAN and RA, is required for robust SAN function and protection from SAN arrest during adenosine challenge. The heterogeneity in the expression of A1R within the human SAN compartments underlies the direction of pacemaker shift and preferential SACPs in the presence of adenosine. Alterations of $I_{Na}$ current and fibrotic remodelling in SACPs can significantly modulate SAN conduction and shift the preferential SACP/exit from SAN. Finally, we show that disease-induced fibrotic remodeling, $I_{Na}$ suppression or increased adenosine make the human SAN vulnerable to pacing-induced exit blocks and reentrant arrhythmia. In summary, our computer model recapitulates the structural and functional features of the human SAN and can be a valuable tool for investigating mechanisms of SAN automaticity and conduction as well as SAN arrhythmia mechanisms under different pathophysiological conditions.

we used in this study are available in the CellML depository. The human SAN cellular model can be downloaded from Physiome CellML repository: https://models.physiomeproject.org/workspace/648. The human atrial cellular model is accessible through https://models.physiomeproject.org/e/807/ni_2017.cellml/view. In our study, we applied and adapted the original cellular SAN models for different SAN regions with SAN regional heterogeneity to qualitatively match experimental results. A custom-built C software was used to generate all computer simulations and has been described previously by our group [DOI: 10.1038/s41467-019-14039-8 and DOI: 10.1161/JAHA.117.005922]. Custom C code and related files used for computer simulations and the human SAN model in this study have been deposited at GitHub: https://github.com/rsha919/Human_SAN_2D_Fabbri_Ni.

Funding: This work was supported by NIH HL115580 and HL135109 (VVF), the Bob and Corrine Frick Center for Heart Failure and Arrhythmia, the Ohio State University, the Health Research Council of New Zealand (#21/355), Royal Society Te Apārangi Catalyst Fund and the National Heart Foundation of New Zealand (JZ). The funders had no role in study design, data collection and analysis, decision to publish, or preparation of the manuscript.

Competing interests: The authors have declared that no competing interests exist.

## Author summary

The human heart is driven and modulated by the sinoatrial node (SAN), our body's natural pacemaker. Recent studies using explanted human hearts discovered that to regulate heart rhythm robustly, the SAN has three pacemaker compartments–SAN head, center and tail characterized by heterogeneous expression of key ion channels and receptors. In addition, the fibrotically-insulated SAN electrically connects with the right atrium (RA) through up to five sinoatrial conduction pathways (SACPs). Due to the complexities of the human 3D structure and limited functional data on SAN conduction, the specific role of the SAN insulation/border, distinct SACPs and intranodal pacemaker molecular heterogeneity in regulating sinus rhythm in health and diseased hearts remain debatable. The goals of this study were to define the key factors influencing human SAN pacemaking function and SAN dysfunction by developing and utilizing computer models of the human SAN. This study presents the first comprehensive biophysical computer model of the human SAN complex based on direct molecular, structural and functional studies in the ex-vivo human heart. Our data show that the computer models can closely replicate pacemaking, SAN activation patterns and exit sites/earliest atrial activation through preferential SACPs, as well as physiological changes including the shift of the leading pacemaker in the presence of adenosine reported in the human heart ex-vivo. More importantly, the novel computer modeling simulation results illustrate the crucial role of the structural and electrical heterogeneity of the human SAN in pacemaking and conduction. Our human-specific SAN computer model represents a valuable tool for investigating SAN automaticity, conduction and arrhythmia mechanisms under normal and various disease conditions.

## Introduction

The sinoatrial node (SAN) is the primary pacemaker of the human heart, responsible for generating and efficiently regulating cardiac rhythm under various physiological conditions [1,2]. The human SAN is a single, "banana-shaped" 3D heterogeneous multicellular structure, composed of specialized pacemaker cells, adipose cells, immune cells, nerve fibers and importantly, ~35–50% dense connective tissue [3–7]. This structure is further compartmentalized into head, central and tail intranodal pacemakers characterized by heterogeneous ion channels and proteins that maintain pacemaking and conduction [8,9]. The distinctive fibrotic tissue in the human SAN together with fatty tissue, and low electrical coupling between pacemaker cells and atrial myocardium along the SAN border create electrical insulations of the intranodal pacemakers from the surrounding right atrial (RA) myocardium. This insulation may facilitate intranodal pacemaking and conduction as well as overcome the sink-source mismatch between the large RA myocardium (sink) and relatively small SAN pacemakers (source) as shown in the pioneer modelling study by Joyner and van Capelle [10]. Additionally, cardiac diseases including heart failure (HF) may lead to pathological molecular and structural (e.g., increased fibrosis) remodeling within the SAN pacemaker complex, resulting in SAN pacemaker and conduction dysfunction (SND) and reentrant arrhythmias [1,2,9,11].

The mechanisms of SAN function and dysfunction have been extensively studied in animal models [12,13]. These animal studies established a foundation of theories on heart rate regulation and possible mechanisms of SND or sick sinus syndrome but also highlighted large interspecies variations [14]. Recent ex-vivo studies reveal that the human SAN complex may be

unique in both structural and electrophysiological aspects that limit the translational applications of animal studies to clinics [1,2,5]. Importantly, these studies revealed that the multiple SAN conduction pathways (SACPs) and intranodal pacemakers within the unique 3D SAN fibrotic structure may provide fail-safe mechanisms to ensure robust, uninterrupted SAN pacemaking and conduction [1,2,5,6]. Yet, due to the complexities of the human 3D structure and limited functional data on SAN conduction, the specific role of the SAN insulation/border, distinct SACPs and intranodal pacemaker molecular heterogeneity in the regulation of sinus rhythm in health and diseased hearts remains debatable. Thus, despite over a century of research on the SAN, limited knowledge of the relationship between human SAN function and the 3D structural-molecular microarchitecture of the human SAN pacemaker-conduction complex remains a critical barrier to properly understanding SAN dysfunction and arrhythmia mechanisms and developing new therapeutic approaches, e.g., biological pacemakers [15].

Computer models of cardiac electrical activation provide a powerful framework for understanding the structural and functional mechanisms of underlying variable physiological and pathological conditions, such as HF. Computer simulations also provide unique opportunities to test the role of each individual factor (e.g. region-specific fibrosis or pacemaker ion channel expression) in SAN pacemaking and conduction functions. These would be impossible to achieve in experimental or clinical studies [9,16]. However, currently existing computer models of the human SAN are either single pacemaker cell models [17] or 2D and 3D models, which do not incorporate human SAN-specific structural/functional/molecular data [18–20]. For these reasons, heart-specific computer models of the human SAN, incorporating molecular, structural and functional data from the same cohort of ex-vivo human hearts [1,5,6,8,9,21] may provide a powerful means to test novel hypotheses.

The goal of this study was to define the key factors influencing human SAN pacemaking function and SAN dysfunction by developing and utilizing computer models of the human SAN complex. This novel SAN *in silico* model was based on data from our recent high-resolution near-infrared optical mapping, molecular, and detailed 3D histological imaging studies directly in the human heart *ex-vivo* [1,9]. Biophysics-based computer models of the human SAN were designed to simulate electrical pacemaking and conduction between SAN, SACPs, and RA based on realistic geometric loading and compartment-specific heterogeneity of molecular and ion channel expressions, as well as the impact of autonomic stimulation with adenosine, HF-induced remodeling and atrial pacing on SAN function.

## Methods

### Human SAN optical mapping and 3D reconstruction

Near-infrared optical mapping data and histological imaging and reconstruction of human SAN used for the current SAN model was published previously in Li et al. 2017 and 2020[5,9] and described in S1 Text. Briefly, *ex-vivo* optical mapped donor human SAN preparations were histologically dissected for 3D structural reconstruction and analysis. 400 histology sections were imaged at a spatial resolution of 0.5×0.5 μm$^2$ using a 20X digital slide scanner (Aperio ScanScope XT, Leica). The high-resolution histology images of the human SAN pacemaker complex were sequentially stacked, and artificial deformation across the z-axis was minimized using a novel 3D image alignment approach [5]. Subsequently, segmentation was performed on the stacks of Masson's trichrome to separate the SAN from the neighbouring RA based on functional and structural data. Myocardial tissue was delineated from fat, blood vessels and fibrosis based on the colour intensity within the 3D SAN complex (**Fig A in S1 Text**). In addition, five SACPs (yellow color) were identified as 1–3 mm of myofibers with

transitional cells in the SAN border that merged with RA myofibers [5]. High-resolution fiber fields were obtained using eigenanalysis of the structure tensor [16].

## Human SAN computer model

Based on the 3D reconstruction of the optically mapped human SAN complex, we developed a SAN-SACP-RA model to conduct computer simulations (**Fig 1**). The reconstructed 3D SAN-SACP-RA anatomical model had a size of 19.5x4.0x2.6 mm$^3$ at an isotropic resolution of 40 μm$^3$. The SAN computer model was obtained using a shadow of the 3D SAN model to the XY plane (parallel to epicardium) as shown in **Fig B in S1 Text**. As a result, the 2D representation of the entire 3D human SAN structure included all SACPs and the complete SAN head/center/tail, which is crucial for the aims of this study. In addition, the computer model used the myofiber field from histology data. Such model reproduced the geometry of electrical connections between the SAN and the neighbouring RA and was much more efficient to run than a computer model of the SAN directly based on the 3D histological data. We did not incorporate the SAN's internal blood vessels into the model (as physical barriers) as they do not affect SAN and RA interaction. The insulating wall (at a uniform thickness of 3 pixels) between the SAN and RA was given a constant potential of -62.5 mV, which is the mean of the resting potentials of the RA and SAN cells, and a 0.001% diffusivity of the RA diffusivity [22–24].

The cellular activation models for the human SAN center/head/tail and SACPs were adapted from the Fabbri et al. 2017 model [17], which is the most widely used human SAN model based on recent experimental data. The following modifications were made by considering SAN regional heterogeneity data from recent studies [1,2,9] (**Fig 1**). The ratios of $I_{Na}$, $I_f$ and $I_{K1}$ currents among the four SAN regions were listed in **Table A in S1 Text**, respectively [1]. The simulated SCLs for isolated SAN pacemakers in the SAN center and head/tail were 813 ms and 798 ms, respectively. The SACP cell models were not able to pace themselves. The baseline condition was considered to be without adenosine. In our computer model of the human SAN complex, the maximum concentration of acetylcholine (ACh), 60 nM, led to SAN

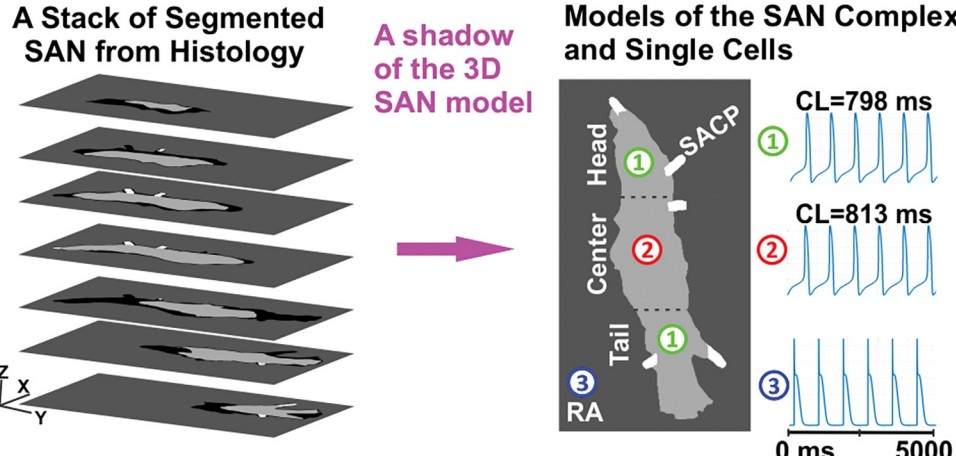

**Fig 1. The reconstruction of the human SAN anatomical and computer activation model.** A representative human SAN computer model at an isotropic resolution of 40 μm$^3$ was constructed using a shadow of the 3D immunehistological segmentation of the human SAN structure including SACPs [1] into a 2D plane. For this computer model, three different cellular kinetics models were developed for SAN center (1), head and tail (2), and neighboring RA (3). SAN–sinoatrial node, RA–right atrium, SACP–sino-atrial conduction pathway, CL–cycle length.

arrest, which we referred to as 100% adenosine and a utilized dose of adenosine was represented as a percentage relative to this maximum value throughout this modeling study. To incorporate the effects of adenosine/ACh into the SAN cellular models, we utilized the same approach as described in the study by Fabbri et al.[17] for modeling the effects of ACh by modulating its concentration. The administration of ACh in the SAN activated the ACh-activated $K^+$ current ($I_{KACh}$), influencing $I_f$, $I_{CaL}$ and sarcoplasmic reticulum $Ca^{2+}$ uptake. In addition, expression of the $I_{KACh}$ channel or A1 adenosine receptor (A1R) was modeled higher in the SAN center than in the SAN head/tail compartments based on data from human experimental studies [1]. Modelling of the relative expression of A1R in the SAN head/tail was achieved by changing the density (max conductance) of the $I_{KACh}$ current at the head and the tail of SAN. The RA cells were modeled by using the recently adapted human atrial Courtemanche et al. cell model [25]. In addition, the impact of adenosine/ACh on RA cells was modeled using the same formula by Grandi et al. [26] To simulate the electrical remodeling in the SAN complex under HF, we introduced $I_f$ and $I_{Na}$ current block by 20% in the SAN and SACPs and 5% $I_{Na}$ block in the RA [9]. To simulate the impact of fibrosis in HF, we used 20% fibrosis in both SAN and SACP regions as we have done previously [9] (**Table B in S1 Text**).

Electrical conduction among SAN pacemakers and RA cells was modeled using a monodomain equation and solved using a parallelized finite difference approach. We used a spatial step of 0.04 mm and a temporal step of 0.0025 ms in our solver. A forward Euler method was used to solve the ordinary differential equations of cellular models. The electrical conduction attributable to intercellular electric coupling via gap junctions was simulated through the diffusion coefficient. In the model, we considered the regional differences in gap junctional coupling between the SAN center, SAN head/tail, SACPs and RA tissues by setting the diffusion coefficients at a ratio of 7:10:20:50 in these regions (**Table C in S1 Text**) [9]. In the models, an anisotropic diffusivity ratio of 1:10 was used as conducted in the past [9,16].

## Results

### SAN activation at baseline and with application of adenosine

The developed control human SAN computer model reliably reproduced SAN rhythm or sinus cycle length (SCL), leading pacemaker location and electrical propagation pattern including preferential SACPs within the SAN complex at baseline and during adenosine administration, which were identical to these parameters recorded during our ex-vivo mapping experiment in the same ex-vivo human heart (**Fig 2**). Under baseline conditions in the computer model (**Fig 2A left**), the leading pacemaker was located in the SAN center (circle), and the earliest RA activation site was through the middle lateral SACP (magenta asterisk). The activation time within the SAN before exiting through SACP was ~75 ms (SAN conduction time—SACT).

Qualitatively similar results were observed in the same human heart optically mapped ex-vivo at baseline conditions (**Fig 2B left**). In the presence of a higher concentration of adenosine (84% of the maximum dose) in the computer model, the leading pacemaker shifted to the SAN tail (**Fig 2 right**). In addition, the earliest RA activation site was activated through the superior lateral SACP. The SACT within the SAN was prolonged to 318 ms. An example of qualitatively similar results observed in the experiment in the presence of a high dose of adenosine is shown in **Fig 2B right**. In that case, under application of 10 μM of adenosine, we also observed the shift of the leading pacemaker to the tail, RA activation through the superior lateral SACP and SACT of 370 ms, which is comparable to the model (318 ms). The activation path of the electrical wave was the same in the model and in the experiment.

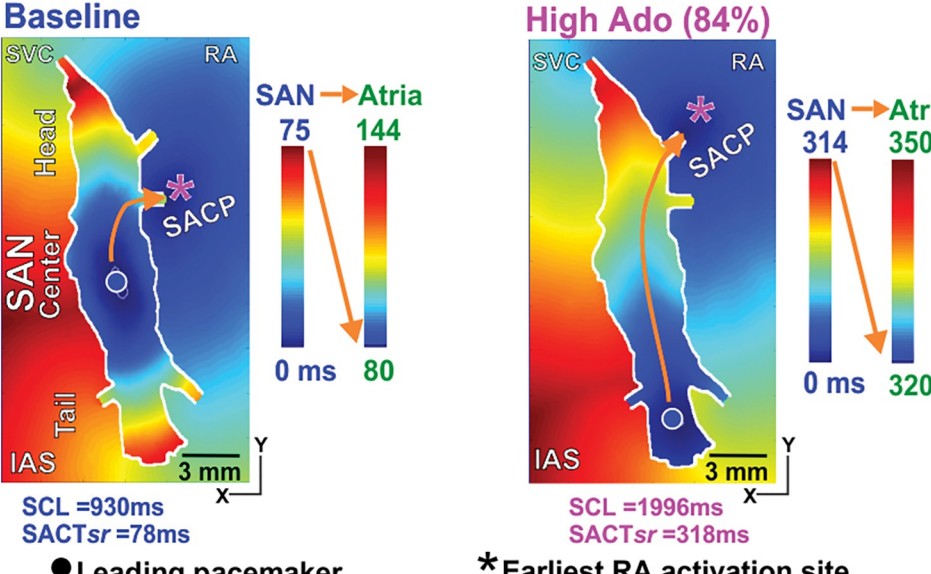

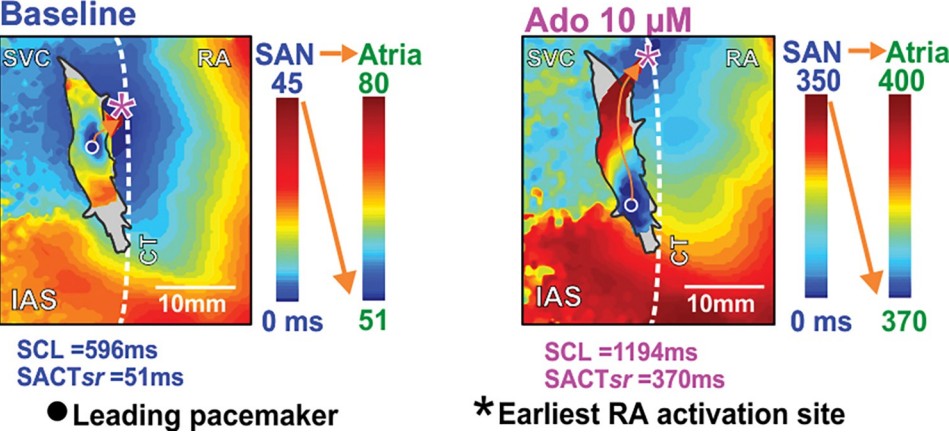

**Fig 2. The computer activation model of the human SAN complex was validated using the optical mapping of the same human heart under both baseline and adenosine (Ado). A,** The activation maps of the human SAN-RA in the computer model where the earliest pacemaker originated from the SAN center and propagated into the RA via the middle lateral SACP under baseline conditions. The introduction of Ado shifted the leading pacemaker to the SAN tail and changed the exit site to the superior lateral SACP as in ex-vivo optical mapping experiments. **B,** Optical mapping of the same human heart ex-vivo had qualitatively similar leading pacemaker sites and propagation patterns [1]. SAN–sino-atrial node, RA–right atrium, SACP–SAN conduction pathway, SACTsr–SAN conduction time during sinus rhythm, Ado–Adenosine, CT–crista terminals, IAS–interatrial septum, SVC–superior vena cava.

Importantly, increasing the concentration of adenosine from 0% to 100% in the computer models (**Figs 2A and 3A**) led to progressive slowing of the SCL and SACT in parallel with a shift in the leading pacemaker and earliest atrial activation sites, until complete atrial arrest at 100% adenosine concentration. The simulation results show that the leading pacemaker shifted inferiorly from the SAN center to the SAN tail, while the earliest RA activation site first shifted to the inferior lateral SACP for adenosine concentration 50% (Fig 3A) and then to superior lateral SACP at adenosine concentration 84% (Fig 2A). Both SCL and SACTsr were

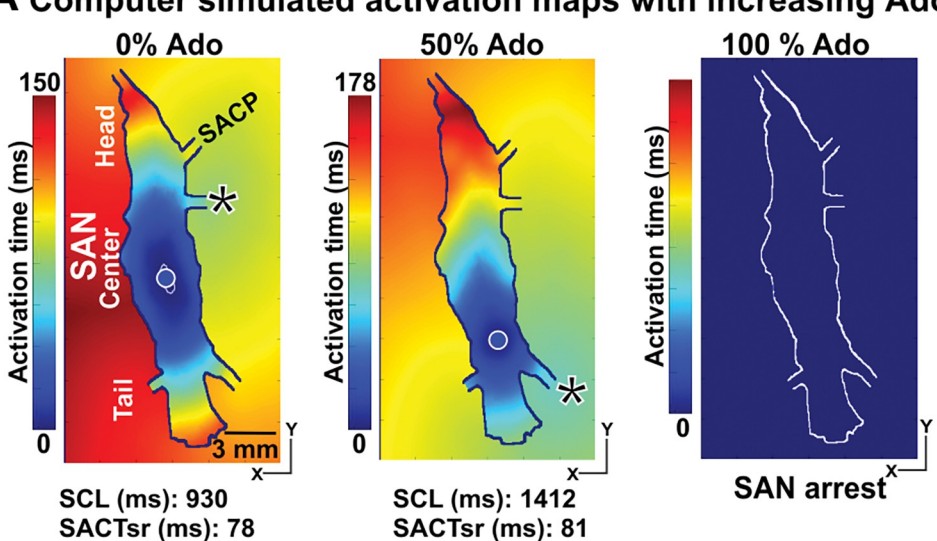

**Fig 3. A shift in the leading pacemaker and earliest atrial activation sites in the human SAN model with varying Ado concentration. A,** Increasing the presence of Ado (from 0% to 100%) in the computer model of the human SAN complex led to a shift in the leading pacemaker and earliest atrial activation sites, eventually exit block and complete atrial arrest (100% of Ado). **B,** A similar shift in the leading pacemaker and earliest atrial activation sites was observed in 11 optically mapped human hearts ex-vivo in the absence and presence of Ado [1]. The increasing dose of Ado led to a heart-specific pacemaker shift toward the head or tail of the SAN complex, and a higher chance of conducting via superior or inferior lateral SACPs as in computer model. Also, 100 μM Ado led to cardiac or SAN arrest in five out of the 11 hearts. SAN–sino-atrial node, SCL–sinus cycle length, SACP–sino-atrial conduction pathway, SACTsr–sinoatrial conduction time during sinus rhythm, Ado–Adenosine.

increased in the computer model with an increasing dose of adenosine. Experimental results in the optically mapped human hearts *ex-vivo* (n = 11) at baseline and in the presence of low (10 μM) and high (100 μM) concentrations of adenosine are shown in **Fig 3B**. In functionally mapped explanted human SANs (n = 11), the increasing dose of adenosine led to a heart-

specific pacemaker shift toward the head or tail of the SAN complex. An increase in concentrations of adenosine led to a higher chance of conducting via superior or inferior lateral SACPs as in the computer model. Also, 100 μM Ado led to cardiac or SAN arrest in five out of the 11 hearts similar to what we observed in our simulations.

## The heterogeneity in expression of A1 adenosine receptors or $I_{KACh}$ channel may explain pacemaker shifts

The heterogeneity in expressions of A1R or $I_{KACh}$ channel in the human SAN ex-vivo was shown in our previous experimental studies [1] (**Fig 4A**). To understand the effect of this heterogeneity on SAN function, we implemented this heterogeneity in A1R[1] and hyperpolarization-activated cyclic nucleotide-gated channel subunits (HCN)[8] to our model with higher expression levels in the SAN center than its head/tail (**Fig 4B**). We performed a series of simulations of SAN activation patterns in which we increased the ratios of expressions of A1R (SAN head/tail to center) from 0.1 to 0.9 in the presence of 20% adenosine. We found that increasing heterogeneity resulted in gradual shift of the leading pacemaker from the SAN center to the tail (**Figs 4C** and **C in S1 Text**).

We also modelled the superior-inferior gradient by changing the A1R expressions in SAN head vs tail (**Fig 4D and 4E**). We performed simulations in the presence of 20% adenosine where the expression of A1R in the tail was change from 0.1 to 0.9, and expression in the head and center were constant at 0.1 and 1.0, respectively (**Fig D in S1 Text**). **Fig 4D** shows two different scenarios of A1R expressions in SAN head vs tail: when A1R expressions is higher in the SAN tail vs head (0.9:0.1), the application of adenosine slowed SAN automaticity and shifted the leading pacemaker from SAN center (baseline condition) to head (superior). The pacemaker shift had opposite directionality (from center to tail) when SAN head and tail have the same A1R expression (0.1:0.1), despite of similar automaticity slowing (SCL from 930ms to 1229 ms vs 1134ms). We documented both these scenarios in ex-vivo human donor hearts studied with near-infrared transmural optical mapping as it shown in **Fig 3B** [1]. There is no leading pacemaker shift with homogenous A1R expressions at the SAN center, head and tail (**Fig E in S1 Text**). Thus in human hearts, the heterogeneity in the expression of A1R within the SAN pacemaker compartments (center, head and tail) could explain the shift of the leading pacemaker and earliest atrial activation sites under adenosine conditions.

Interestingly, under similar conditions of adenosine, almost all leading pacemakers were always in the SAN tail/center and never in the SAN head. We hypothesized that it was due to the regional source-sink relationship within the SAN complex. The SAN head in this specific human heart had fewer SAN cells (electrical sources) and more SACPs (electrical loading) than the tail (three SACPs versus two SACPs), which made excitation more difficult. To test this hypothesis, we performed additional simulations in which we artificially blocked two lateral SACPs in the SAN head (see the two black arrows in **Fig 4E** right), thereby reducing the electrotonic load. As a result, the leading pacemaker shifted from the tail to the head in the presence of 20% adenosine.

## The characteristics of SACPs dictate the earliest atrial activation sites

One of the results of pathological remodelling of cardiac tissue is a change in $I_{Na}$ in the atrial myocardium [9]. We studied how the change in $I_{Na}$ affects the functioning of the SAN complex. We demonstrated that $I_{Na}$ in SACPs is one of the main determinants of the earliest atrial activation sites (**Fig 5A**). Varying the density of the $I_{Na}$ current in the SACPs alone from its original value to 85% or 115% led to the slowing or acceleration of SACT, and the earliest atrial activation shift from the middle lateral SACP to the superior and inferior lateral SACPs or the

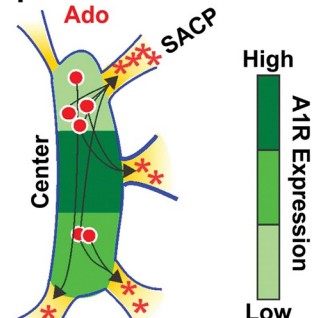

**A** SAN center has higher A1R/GIRK expressions than head/tail

● Leading pacemaker   ✱ Earliest atrial activation site

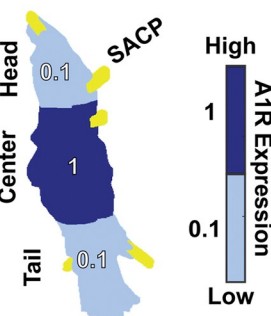

**B** Higher A1R expressions in the SAN center (10:1) was used in the model

**C** Simulated activation patterns in the SAN in the presence of 20% Ado and gradually increasing A1R in the SAN head & tail while keeping center constant

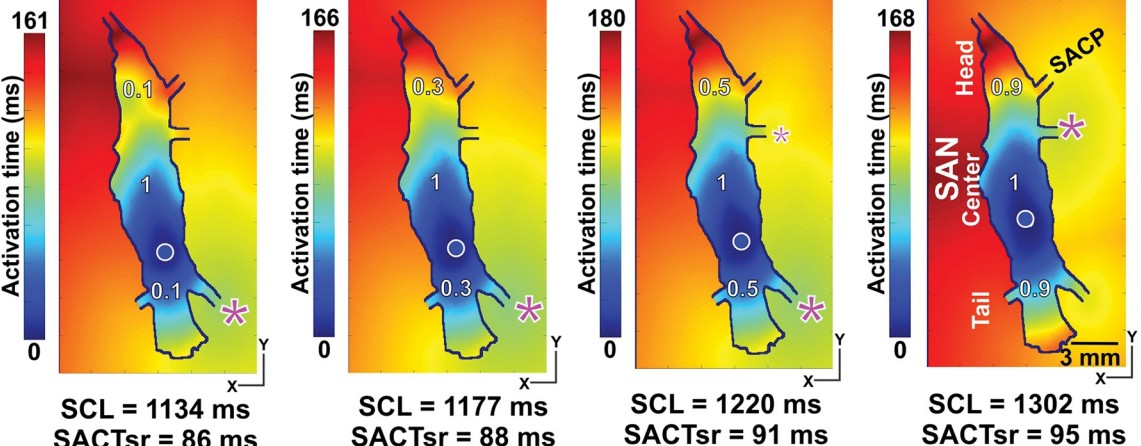

SCL = 1134 ms
SACTsr = 86 ms

SCL = 1177 ms
SACTsr = 88 ms

SCL = 1220 ms
SACTsr = 91 ms

SCL = 1302 ms
SACTsr = 95 ms

**D** Activation patterns at 20% Ado with varying A1R in the SAN tail

SCL = 1134 ms
SACTsr = 86 ms

SCL = 1229 ms
SACTsr = 74 ms

**E** Impact of source-sink relationship on activation by blocking two SACPs

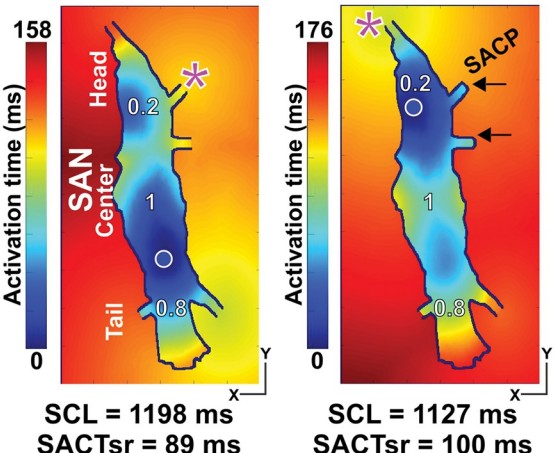

SCL = 1198 ms
SACTsr = 89 ms

SCL = 1127 ms
SACTsr = 100 ms

**Fig 4. The heterogeneity in expressions of A1R and regional source-sink ratios within the SAN complex explains leading pacemaker shifting.** The higher expressions of A1R in the SAN center than that in the SAN head/tail was suggested in the human SAN ex-vivo [1] (**A**) and used in the computer model (**B**). **C,** Increasing expressions of A1R/$I_{KACh}$ channel in the SAN head/tail relative to the center (from 0.1 to 0.9) led to a shift in the leading pacemaker. **D,** Increasing the A1R expressions in the SAN tail (from 0.1 to 0.9) led to the leading pacemaker in the SAN head. **E,** Blocking two right SACPs (see black arrows) in the SAN head reduced electric sink/loading, shifting the leading pacemaker in the SAN head with 20% Ado. A1R –A1 Ado receptors. See other abbreviations in **Fig 2**.

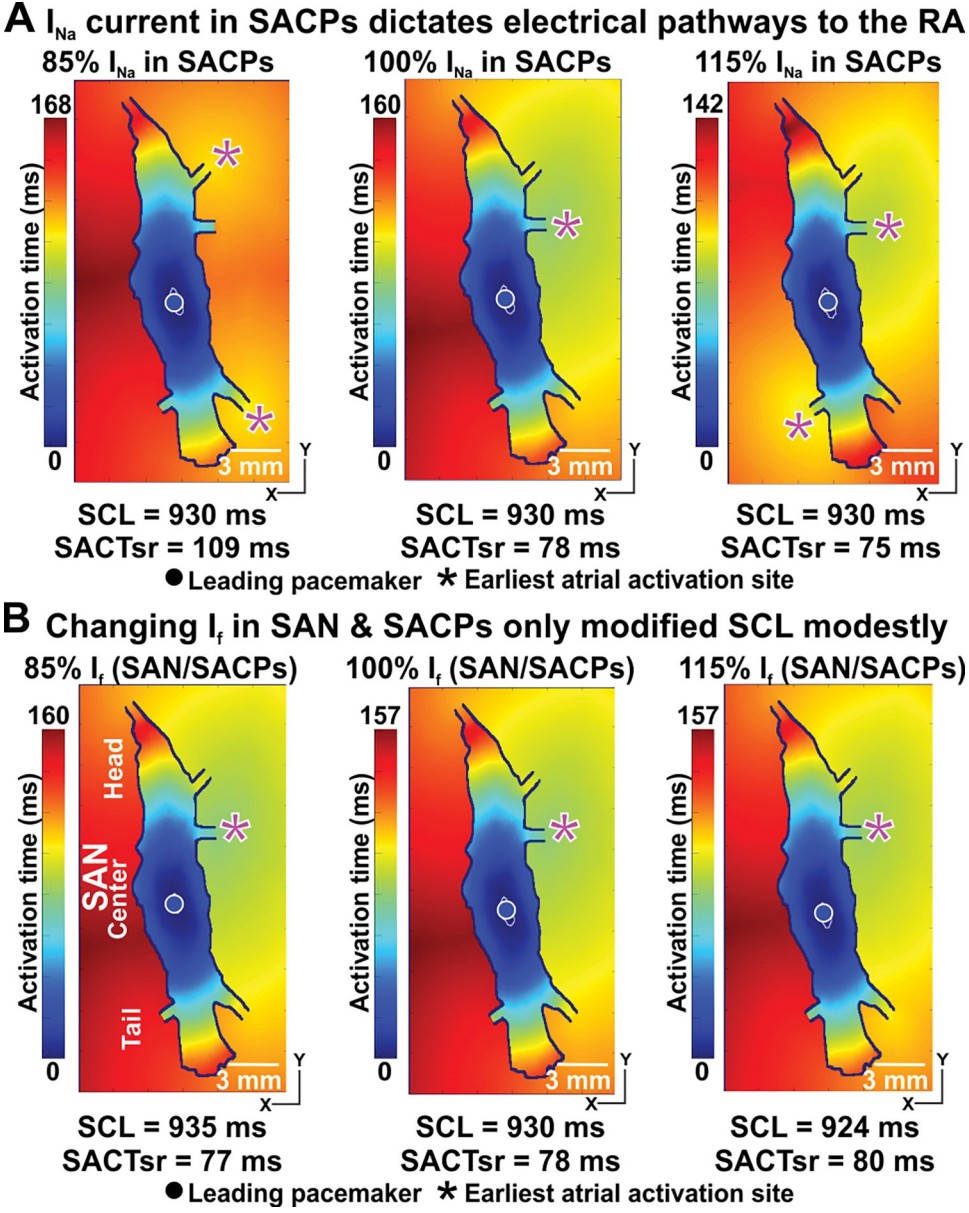

**Fig 5. Modulation of $I_{Na}$ current in SACPs indicates the earliest atrial activation sites, while $I_f$ current only influences SCL. A,** Varying the $I_{Na}$ current in SACPs alone heavily influenced the earliest atrial activation sites. **B,** Varying the $I_f$ current in SAN and SACPs did not change the earliest atrial activation sites. Both $I_{Na}$ and $I_f$ currents were positively associated with the heart rate. See abbreviations in **Fig 2**.

middle lateral and the left inferior SACPs, respectively. However, the SCL and leading pacemaker remained unchanged (Fig 5A). Both $I_{Na}$ and $I_f$ currents within the SAN complex were inversely associated with SCL. However, it appeared that $I_{Na}$ and $I_f$ currents in the SAN did not influence the earliest atrial activation or the leading pacemaker sites (**Fig 5B**).

## The role of the insulation boundary between the SAN and RA

The existence of an insulation boundary between the SAN and the RA septum is widely observed and accepted. However, the existence of the insulation boundary between the SAN

and the lateral RA is controversial. To illustrate the necessity and potential role of the insulation boundary between the SAN and the lateral RA, we performed computer simulations with and without the insulation boundary (**Fig 6A**). In the case of no insulation boundary, we considered situations with different diffusivities within the SAN complex (the diffusion coefficient is 100%, 50% and 25% of its normal value). Reducing diffusivities within the SAN complex led to increased heart rate and prolonged SACT within the SAN. We found that computer models with 100% and 50% diffusivities produced realistic activation time between 150–200 msec. Using computer models without the insulation boundary between the SAN and the lateral RA, we observed only a mild shift of the leading pacemaker and the earliest atrial activation sites with increasing adenosine (**Fig 6B and 6C**), which is not consistent with that commonly seen during experimental studies of animal and human hearts [1,2,9]. Furthermore, even 10% Ado was sufficient to induce SAN arrest (**Fig 6B**), which is 10 times lower than that required to induce SAN arrest with an insulation border (**Fig 3A**). Thus our simulations suggest that the insulation boundary between the SAN and the lateral RA is necessary for normal functioning of SAN and support findings from experimental studies with adenosine.

## The impact of HF-induced remodeling on SAN pacemaking and conduction

We have also studied the effects of ion channel remodeling and fibrosis due to HF on SAN complex function. We implemented these changes to our model as shown in (**Fig 7A**). This figure also shows that electrical remodeling in the SAN cellular kinetics models led to depression of SAN pacemaking and increased SCL. In our computer simulations, SAN with HF conditions led to exit block even without the presence of adenosine (**Fig 7B**). When we reversed the fibrotic remodeling only in the SACPs (**Fig 7C**), we observed electrical activation and propagation in the RA at the baseline and in the presence of up to 50% of adenosine. We also found a similar trend in shifting leading pacemakers and earliest atrial activation sites. At 85% of adenosine, electrical remodeling without fibrosis in SACP led to exit block (**Fig 7C right**). We also found that the fibrotic remodeling in the SACPs alone (20% of fibrosis) without electrical remodeling, produced exit block similar to that under HF remodeling. On the other hand, in the computer model with HF ion channel remodeling only (**Fig 7D**), we again observed electrical activation and propagation in the RA from the baseline to the presence of 50% adenosine. At 85% adenosine, it led to SAN arrest. Our simulation results indicated that HF ionic channel remodeling influenced the SCL, earliest atrial activation sites and increased chances of SAN arrest, while fibrotic remodeling in SACPs increased the chance of SAN exit block.

## SAN is prone to arrhythmia and exit block under $I_{Na}$ channel block, adenosine and HF

Finally, we evaluated the human SAN function after RA pacing (**Figs 8** and **F in S1 Text**) and without any RA pacing (**Fig G in S1 Text**). In the computer model of the human SAN complex, a train of stimuli at a pacing cycle length of 500 ms was delivered from the right superior RA, and a typical activation pattern is shown in **Fig 8A**. Under baseline conditions, once the RA stimuli were terminated, the SAN recovered to its normal automaticity and function immediately. We performed systematic simulations in which we varied the degree of the $I_{Na}$ channel block and concentration of adenosine. We found that depending on these parameters we can observe the following situations: a shift in the leading pacemaker and the earliest atrial activation sites, SAN-RA reentry, SAN exit block and arrest (**Fig 8B**). Typical propagation patterns for each of the cases are shown in **Figs 8B** and **F in S1 Text**. RA pacing led to both

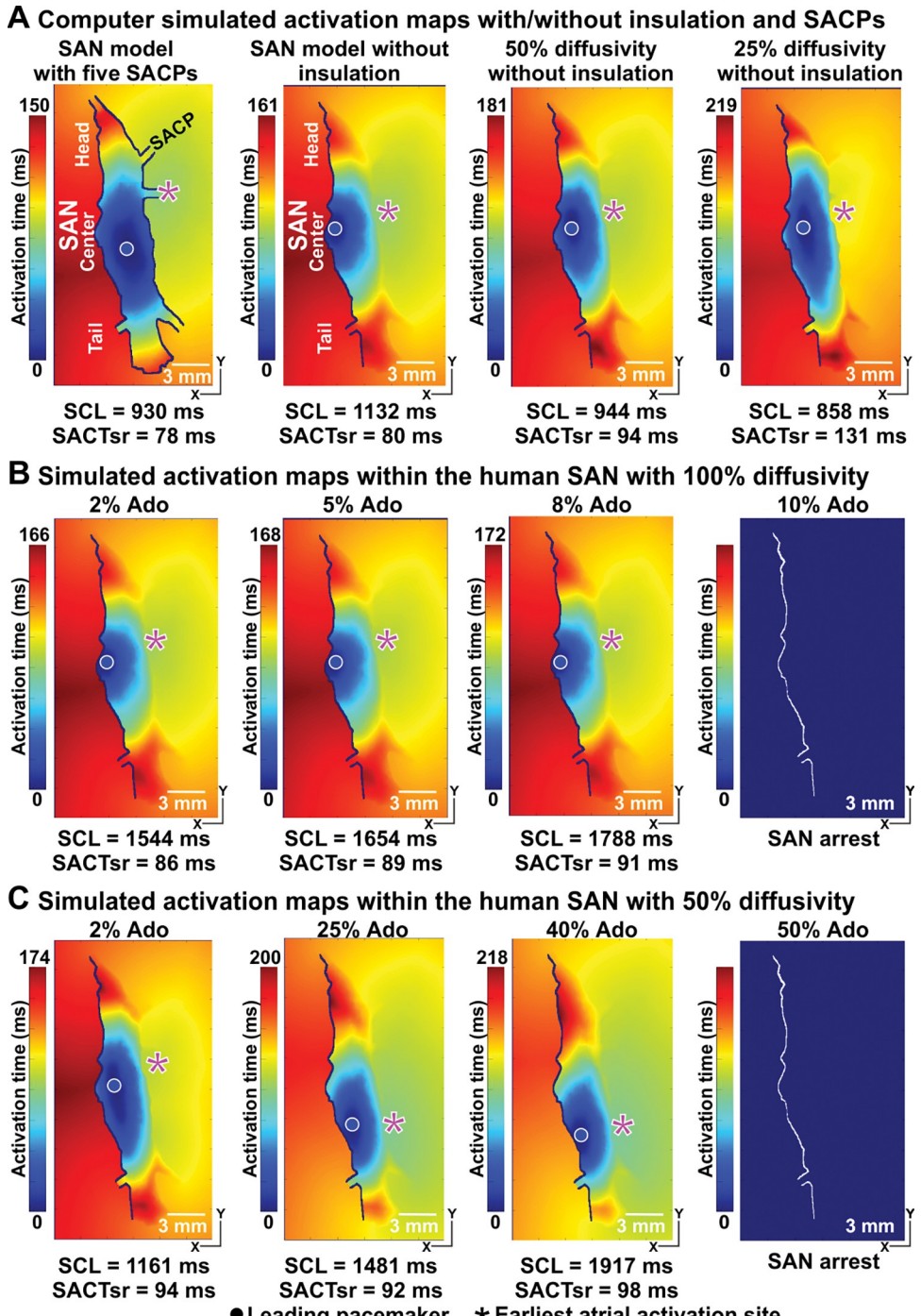

**Fig 6. The human SAN model without the insulation boundary between the SAN and the lateral RA (right) had difficulty in replicating dramatic shift in the leading pacemaker and earliest atrial activation sites. A,** Activation patterns and earliest atrial activation sites in the computer model of the human SAN complex, and in the computer models of the SAN without the crista terminalis (CT) side insulation boundary at different conduction diffusivities. **B and C,** Activation maps in the computer models of the SAN without the insulation boundary at 100% conduction diffusivity and at 50% conduction diffusivity at the baseline and with increasing the concentrations of Ado up to 10% **(B)** and 50% **(C)** which led to SAN arrests, respectively. See other abbreviations in **Fig 2**.

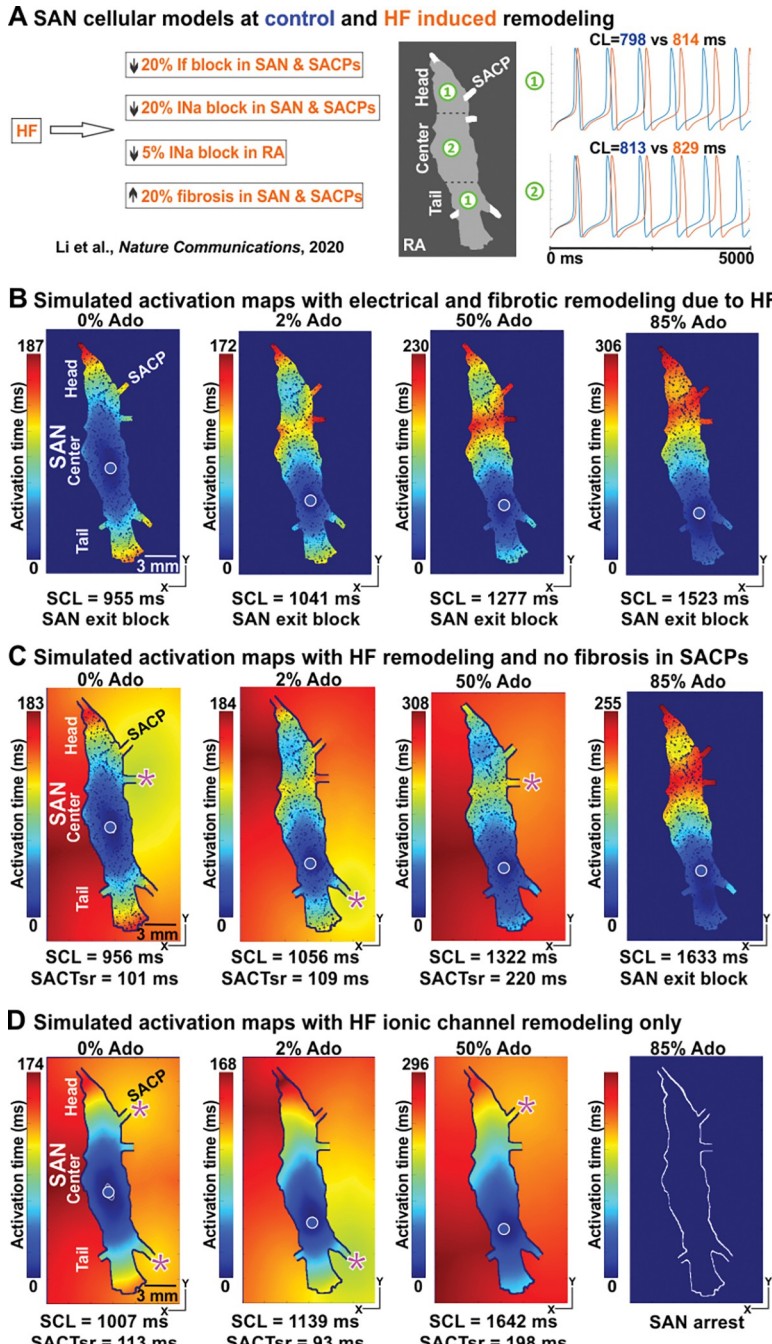

**Fig 7. Modeling of diseased induced fibrotic and ion channel remodeling in SAN and SACPs at the baseline and with Ado. A,** SAN and RA cellular models were further adapted by varying $I_f$ and $I_{Na}$ currents and 20% fibrosis were added to the model to simulate HF conditions. HF-induced electrical remodeling slowed the heart rate. **B,** In our computer simulations, SAN with HF conditions led to exit block even without the presence of Ado. **C,** When we reversed the fibrotic remodeling in the SACPs only, we observed electrical activation and propagation in the RA from the baseline and in the presence of up to 50% of Ado. **D,** With HF ionic channel remodeling only, electrical activation and propagation in the RA were observed from the baseline to the presence of 50% Ado. While 85% Ado led to SAN arrest. HF–heart failure, SCL–sinus cycle length, see other abbreviations in **Fig 2**.

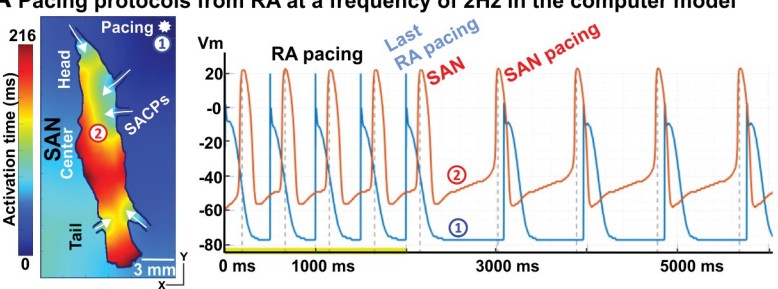

**A** Pacing protocols from RA at a frequency of 2Hz in the computer model

**B** Summary of propagation patterns in the control SAN complex post RA pacing

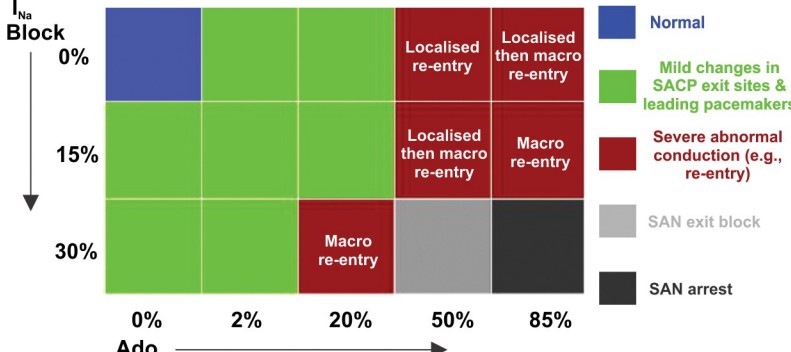

Typical propagation patterns within the SAN complex post the RA pacing

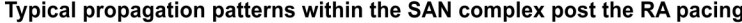

**C** Summary in HF SAN without SACP fibrosis in presence of $I_{Na}$ block or Ado

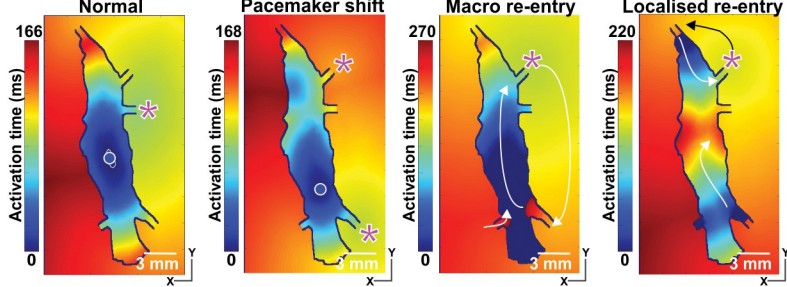

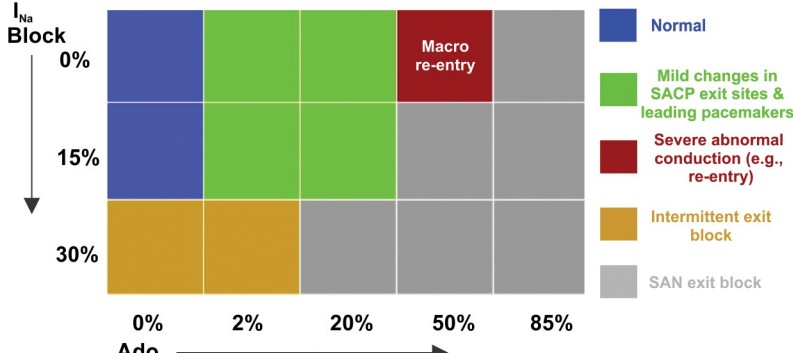

**Fig 8. Atrial pacing induces SAN nodal reentrant arrhythmia during mild $I_{Na}$ suppression and Adenosine (Ado), and SAN conduction block, particularly in HF models. A,** Atrial pacing protocol at a frequency of 2 Hz for SAN function from the RA. The pacing site and SAN activation maps during RA pacing are shown. **B,** Top: Summary of propagation patterns post RA pacing in control SAN model with increasing $I_{Na}$ block or Ado concentration. Here 'Normal' (blue squares) refers to the leading pacemaker in the SAN center and preferential SAN exit through the middle lateral SACP. Bottom: Examples of post-RA pacing activation patterns in the SAN complex are shown. **C,** Summary of propagation patterns in the HF SAN model without fibrotic remodeling in the SACPs in the presence of increasing $I_{Na}$ block or Ado. See abbreviations in **Fig 2**.

localized and macro-reentries. In contrast, we only observed macro reentries in the control SAN model with increasing $I_{Na}$ block or adenosine without extrastimuli (**Fig G in S1 Text**). The SAN HF model produced SAN exit block post-RA pacing regardless of the presence or absence of $I_{Na}$ block or adenosine. Interestingly, the SAN HF model without fibrotic remodeling in the SACPs had a higher tendency of exit block with increasing $I_{Na}$ block or adenosine (**Fig 8D**). Thus we see that the application of adenosine in combination with $I_{Na}$ channel block leads to cardiac arrhythmias and other serious pathologies.

## Discussion

This study presents the first comprehensive biophysical computer model of the human SAN complex based on direct molecular, structural and functional studies in the ex-vivo human heart. Our data show that the model can closely replicate pacemaking, SAN activation patterns and exit sites /earliest atrial activation through preferential SACP, as well as physiological SACT and SCL changes including the shift of the leading pacemaker in the presence of adenosine reported in the human heart ex-vivo [1,9].

Importantly, the novel modeling simulation results provide mechanistic insight into the crucial role of the structural and electrical heterogeneity of the human SAN in the pacemaking and conduction function. More specifically:

1. The heterogeneity in the expression of adenosine A1 receptors (A1R) or the $I_{KACh}$ channels within the human SAN pacemaker compartments explains leading pacemaker and preferential SACP shifts in the presence of adenosine.

2. The electrical insulation boundary between the SAN and RA except the SACP is required for normal SAN pacemaker and conduction function and to reproduce the leading pacemaker and the earliest atrial activation sites, observed in experimental and clinical studies. Importantly, the insulation prevents the high sensitivity of SAN pacemaking to adenosine, including complete SAN arrest seen at lower doses of adenosine in the models without the insulation boundary.

3. The $I_{Na}$ current density and fibrotic remodelling (e.g. in heart failure HF) in SACPs modulate the SAN conduction (e.g. exit block) and the preferential SACP/exits to the atria (e.g. earliest atrial activation).

4. Intranodal $I_{Na}$ current suppression or low-dose adenosine intervention leads to shifts in the leading pacemaker and the earliest atrial activation sites and renders the human SAN pacemaker-conduction complex vulnerable to SAN-RA reentry, SAN exit block and arrest. The SAN HF model had a higher incidence of exit block regardless of the presence or absence of $I_{Na}$ block or adenosine.

### Structural and electrical heterogeneity of the SAN

Since the discovery of the SAN by Keith and Flack [3] more than one century ago, significant strides in our understanding of the SAN complex have been made [5,6,12,27]. It is widely accepted that the heterogeneous distribution of specialized ion channels, intracellular $Na^+/Ca^{2+}$ handling proteins, gap junction channels and receptors within the human SAN complex are some of the few critical players in SAN pacemaking [5]. In particular, expressions of ion channels (e.g. $Na^+$, $Ca^{2+}$ and HCN1/4), and gap junction proteins are heterogeneous within the SAN pacemaker complex [27,28]. Furthermore, in contrast to other cardiac regions, the human SAN has more extensive fibrosis, which is further upregulated in SAN dysfunction or HF [6].

It is also well established that the existence of a septal area of conduction block, known as the *block zone*, prevents the spread of the electric impulse to the interatrial septum directly from the SAN. The precise mechanism of atrial activation by the leading pacemaker remains a controversy. In the past, the *gradient model*, in which there is a gradual change in the intrinsic properties of pacemaker cells from the center to the periphery of the SAN, was proposed to explain how SAN could effectively pace the atria [12] The intrinsic pacemaker activity is greater in cells from the periphery than from the centre of the SAN [29–33]. However, at the tissue level, the periphery of the SAN is connected to a large mass of atrial muscle in the crista terminalis (CT) through gap junctions resulting in the inhibition of peripheral pacemaker activity by the electrotonic influence of the highly hyperpolarized atrial muscle. On the other hand, central pacemaker cells, which are more distal from the atria, are less affected by atrial electrotonic effects. Therefore, leading pacemaker activity at baseline conditions always originates in the central SAN cells, although they are intrinsically slower than the peripheral pacemaker cells. These results were primarily based on small animals studies, including rabbit models, which show both lateral (towards CT) and superior-inferior gradient of intrinsic SAN pacemaker properties. However, in large animal models including canine and human SAN, the superior-inferior gradient is more prominent while lateral gradient is evident only across the SACPs, due to the larger CT myocardium requiring more lateral insulation [1,5,6,9]. Earlier Crick et al. [34] reported twice higher density of parasympathetic and sympathetic nerves fibers in the SAN center vs periphery (tail). The innervation gradient may explain why sympathetic activation can shift atrial exits to superior SACP and parasympathetic activation can slow sinus rhythm and shift atrial exits inferiorly [35]. However, in both human and canine SAN, the direction of intranodal pacemaker shift from center to head or tail does not always correlate with the closest superior or inferior SACP [1,36]. SAN activation can exit via the superior SACPs even though the leading pacemaker shifted inferiorly to the tail (**Figs 2B and 3B**). These studies suggest that SAN automaticity gradient and superior or inferior intranodal pacemaker shift (from center to head or center to tail) depend on the heart-specific SAN compartment molecular profiles. However, no one yet measure and compare intrinsic frequency of pacemaker cells isolated from different SAN pacemaker compartments (head, center and tail). Instead, we included the electrophysiological and molecular difference between the SAN center and head/tail (periphery), as well as transitional cells in preferential SACP for electrical coupling from SAN to RA. In keeping with the classical gradient model, our modelling results showed that isolated SAN pacemakers in the SAN center were slower than the SAN head/tail, which is consistent with most experimental results [29–33].

The most significant controversy surrounds the location and nature of the SAN in the human heart. Dobrzynski and her colleagues found an intermediate region with an expression of many ion channels between the SAN and RA in the human heart, which is similar to that seen in the periphery of the rabbit SAN [27]. They speculated that this region, termed the *paranodal area*, might contribute to pacemaker shift though they did not conduct any electrophysiological or optical mapping studies to support the claim. Using high-resolution optical mapping and histological studies in the human heart ex-vivo, we have shown that the human SAN complex is a 3D, specialized multi-compartment structure (head, center and tail) with higher expression of HCN and A1R proteins, lower gap junctional coupling and Nav1.5 in the SAN center than in the SAN head and tail [1,9]. Our comprehensive biophysical computer model of the human SAN pacemaker conduction complex was developed based on the functional and structural mapping in the human heart to investigate the role of human SAN structure and ion channels heterogeneity in SAN pacemaking and conduction functions.

## The novelty of our computer model of the human SAN

Computer models provide a powerful tool for the quantitative examination of structural and electrical substrates and their individual contributions to cardiac arrhythmia mechanisms [9,16]. However, much less literature exists for multi-scale human SAN modeling, mainly due to the structural and functional complexity of the human SAN complex. The first computer modeling of the SAN by Joyner and Capelle [10] demonstrated that the electrical uncoupling of the pacemaker cells might be an essential design feature of a healthy SAN complex using a two-dimensional sheet model. Similar approaches were expanded to study the importance of gradients in membrane properties (e.g., the ionic current density of the $I_{Na}$) and electrical coupling in 2D [18] and 3D [19] models, and the mechanisms by which the SCN5A mutations ($Na^+$ channel) impair cardiac pacemaking [37]. Earlier 3D computer models of the human SAN complex were developed primarily for studying the electrical conduction within the RA [4,28]. The more recent 3D computer model by Kharche et al. [20] was the first to study the role of the insulating border between the SAN and RA septum, and the paranodal area in the SAN function. However, they only used the simple three-current Fenton-Karma cellular models to simulate the human SAN and RA cell kinetics. In addition, SACPs in their model were not anatomically based, which led to extremely short un-physiological SAN conduction. To date, there is no comprehensive SAN model that integrated realistic anatomical structures and ion channel expressions from direct studies of the human SAN complex. As such, no computer model could successfully reproduce all functional observations at different conditions in normal and diseased human SAN, including the location of leading pacemakers, preferential SACP exits, physiological SAN conduction time (70–80 ms at baseline) and SCL (700–900 ms).

Our study takes the next step in defining the key factors influencing human SAN pacemaking function and SAN dysfunction by developing and utilizing computer models of the human SAN based on the current knowledge of electrical and structural heterogeneity. The unique strength of our SAN computer model is that it was based on published data from high-resolution near-infrared optical mapping, molecular mapping and detailed 3D histological analyses of the human SAN complex ex-vivo [1,5,8,9]. More importantly, it included anatomically based SACPs directly identified from the 3D immunohistological analyses which provide realistic SAN to RA electric loading. In addition, we have adapted the most widely used human SAN and atrial cellular activation models, i.e., the Fabbri et al. [17] and Courtemanche et al. cell model [25].

## Critical functional insights from the human SAN model

Firstly, we have illustrated the role of structural and electrical heterogeneity in the shift of the leading pacemaker and the earliest atrial activation sites. Under baseline conditions, the leading pacemaker is more likely located in the SAN center due to less electrical loading (the relatively larger SAN center region with only one SACP, in contrast with the SAN head and tail), despite the fact that single isolated SAN head or tail cells have a shorter SCL than that in the center (798 ms versus 814ms). The difference in cellular SCL is caused by the electrical heterogeneity within the SAN complex, particularly, the higher $I_f$ and lower $I_{Na}$ currents in the SAN center, as widely reported. That is to say that the effect of the electrical heterogeneity is suppressed by the SAN structure (SAN-RA isolation layer and SACPs). In the presence of adenosine, the shift of the leading pacemaker from the SAN center to the head or tail is made possible by the higher A1R expression in the center, in addition to the heterogeneity of ion channels $I_{Na}$ and $I_f$ currents between the center and head/tail. Our modeling study also shows that the difference in the source-sink ratio between the SAN head and tail can also influence the new leading pacemaker site. Along with the change in the leading pacemaker, the earliest

atrial activation sites shift in locations as well, though it does not always exit through the nearest SACP to the leading pacemaker.

Secondly, our study sheds new light on the role of the insulation boundary between the SAN and neighbouring RA. It is the first time that a computer model of the 3D human SAN complex with the insulation boundary can replicate the dramatic shift in the leading pacemaker and the earliest atrial activation sites as commonly observed in the high-resolution optical mapping of the human heart ex-vivo and clinical studies [9]. On the other hand, modeling simulations suggest that computer models with reduced diffusivities within the SAN can only produce a modest shift. In addition, the leading pacemaker tends to localize close to the insulation boundary between the SAN and the septum which is not consistent with the reported results in the literature [9]. Therefore, our modeling study lends further support to the existence of the insulation boundary between the SAN and neighbouring RA, except SACPs for connecting RA electrically.

Finally, our computer simulations indicate that the $I_{Na}$ remodeling and fibrosis upregulation in the SACPs play a key role in the shift of the earliest atrial activation sites and in the SAN exit block.

The reduction in $I_{Na}$ current density and fibrotic remodeling slow down conduction is widely accepted and are well-known arrhythmogenic factors [38–41]. However, the novelty of this study is that the $I_{Na}$ current density and fibrotic remodeling in SACPs are more important than in other SAN regions. Therefore, the SACP may be a potential therapeutic target in clinics for patients with SAN dysfunction. For instance, reversing fibrotic remodeling in the SACPs will improve the electrical conduction of the SAN to the heart and alleviate the need for electronic pacemaker implantation. Further development of our modeling analysis for clinical intervention may provide a powerful, safe approach to test novel treatments to treat SND.

## Study limitations

Our computer model of the human SAN complex was based on a shadow of high-resolution histology images by projecting the SAN model to the imaging plane. It had realistic geometric regions proportional to the neighboring RA, and different SAN compartments, so it is not a complete 3D representation of the 3D human SAN complex. Due to the current paucity of human SAN compartment-specific electrophysiological data, we used the same cellular model for SAN periphery compartments (head and tail). A computationally efficient, biophysics-based computer model of the entire 3D human SAN pacemaker-conduction complex and RA directly based on the 3D imaging data is yet to develop and validate the insights learned from this study. However, we suggest that taking into account 3D effects may not conceptually affect the main conclusions drawn from our study as the effects of the SAN structural and molecular features (e.g. A1R and ionic channels) on the superior/inferior shift of the leading pacemaker and preferential SACP/earliest atrial activation sites are confirmed by human SAN experiments.

## Conclusions

Our novel biophysical computer model of a human SAN conduction complex combining ex-vivo functional and 3D structural imaging at the highest resolution to date illustrates for the first time the crucial role of the structural and electrical heterogeneity of the human SAN in the pacemaking function. Particularly, our results lend support to the necessity of the insulation boundary between the SAN and neighbouring RA for robust SAN pacemaker and conduction function. The study also suggests that the cardiac disease or drug modulations of the $I_{Na}$ current and fibrosis in intranodal pacemaker compartments and/or SACPs may promote

SAN reentrant arrhythmias. The further development of 3D computer models based on the human heart ex-vivo may provide a powerful, safe approach for preclinical testing of novel treatment for patients with SAN dysfunction worldwide.

## Supporting information

**S1 Text. Supplemental Materials. Table A**. Relative ratios of the densities of the four key ionic channels ($I_f$, $I_{Na}$, $I_{K1}$ and $I_{KACh}$ currents) among the human SAN head, center and tail, and SACPs used in the computer modeling of the control SAN. SACP- sinoatrial pathways, SAN–sinoatrial node, SCL–sinus cycle length. **Table B**. Relative ratios of the densities of the four key ionic channels: $I_f$, $I_{Na}$, $I_{K1}$ and $I_{KACh}$ (A1R expression), and fibrosis among the human SAN head, center and tail, and SACPs used in the computer modeling of HF. SACP- sinoatrial pathways, HF–heart failure, A1R –A1 adenosine receptor, SAN–sinoatrial node, SCL–sinus cycle length. **Table C**. The regional differences in gap junctional coupling between the SAN center, SAN head/tail, SACPs and RA tissues by setting different diffusion coefficients in these regions. SACP- sinoatrial pathways, SAN–sinoatrial node, RA–right atrium. **Fig A**. **3D microstructural composition of the human SAN complex** (an *ex-vivo* human donor heart,). **A,** 3D microstructure of all tissue types including myofibers, fibrosis, and fat in the SAN complex (red) and surrounding atrial tissue (green). **B,** The myofibers of the SAN complex and surrounding atrial tissue. **C,** The fibrotic fibers of the SAN complex and surrounding atrial tissue. **D,** The fat texture of the SAN complex and surrounding atrial tissue. SAN–sino-atrial node, CT–crista terminals, IAS–interatrial septum, SVC–superior vena cava, RAA–right atrial appendage. **Fig B**. **The structure of human SAN computer model.** The SAN model structure was obtained using a shadow of the 3D SAN reconstruction to the XY plane (parallel to epicardium) by project all 2Ds into one plane. As a result, the 2D representation of the entire 3D human SAN structure included all SACPs and the complete SAN head/center/tail. SAN–sinoatrial node, SACP–sinoatrial pathways. **Fig C. Changes in leading pacemaker locations and SACPs in the presence of 20% adenosine due to increasing A1R expression level in the SAN head and tail from 0.1 to 0.9 while keeping A1R in the SAN center constant as 1.** SACP- sinoatrial pathways, SAN–sinoatrial node, RA–right atrium. **Fig D. Changes in leading pacemaker locations and SACPs in the presence of 20% adenosine due to increasing A1R expression level in the SAN tail from 0.1 to 0.9 while keeping A1R constant in the SAN head and in the center**. A1R was set at 1 to the SAN center and 0.1 to the SAN head. SACP-sinoatrial pathways, SAN–sinoatrial node, RA–right atrium. **Fig E**. **The heterogeneity of adenosine A1 receptors (A1R) expressions or the $I_{KACh}$ channel within the intranodal pacemaker compartments (head, center and tail) of the SAN complex is required for the shift in the leading pacemaker and the earliest atrial activation sites during adenosine**. **A,** For the heterogeneous model, the A1R/$I_{KACh}$ expression in the SAN center is 10 times higher than in head/tail (10:1) was used in the heterogeneous A1R computer model. **B,** In the heterogeneous A1R model, administration of 20% Adenosine led to both the leading pacemaker shift (from center to tail) and the shift of the earliest atrial activation site/ preferential SACP from the lateral to inferior SACP. **C,** In contrast, in the SAN model with homogeneous A1R/$I_{KACh}$ expression, the same 20% Adenosine didn't lead to the leading pacemaker and preferential SACP shifts but more severely suppressed SAN automaticity and conduction. The activation maps were almost identical for homogenous A1R with 20% Ado and baseline. SAN–sinoatrial node, SACTsr–SAN *conduction time during sinus rhythm*, SCL–sinus cycle length. **Fig F**. **Four activation patterns were observed after the cession of right atrial (RA) pacing with a CL of 500 ms in control SAN model** (see **Fig 8B** in the main manuscript). **A,** Normal SAN activation pattern: the first post-pacing SAN beat had the same activation pattern as SAN beats

before pacing with the leading pacemaker in the center and preferential conduction exit through the middle lateral SACP. **B,** Mild changes in pacemaker/SACP post the RA pacing: the first post-pacing SAN beat had different activation compared with pre-pacing SAN activation with both leading pacemaker and SACP exit site shifts. **C,** Severe abnormal conduction: SAN macro reentries with slower intranodal conduction path between inferior and superior SACP and a CL of 451 ms spontaneously occurred after RA pacing. **D,** Severe abnormal conduction–Localized SAN reentry between two superior SACPs induced by RA pacing. Two action potential (AP) tracings are from RA, near the pacing site, and the other is located in the center of the SAN. SAN–sino-atrial node, SACP–SAN conduction pathway, Ado–Adenosine, AP–Action potential, CL–cycle length. **Fig G**. **Summary of propagation patterns in control SAN model with increasing $I_{Na}$ block or Ado** (without extrastimuli). Here control refers to the leading pacemaker in the SAN center and consistent exit through the lateral middle SACP. SAN–sino-atrial node, SACP–SAN conduction pathway, Ado–Adenosine.
(DOCX)

## Acknowledgments

We thank the Lifeline of Ohio Organ Procurement Organization and cardiac transplant surgery department of the Ohio State University Wexner Medical Center for providing the explanted donor hearts, as well as the generous donors and their families whose selfless donations make this lifesaving research possible.

## Author Contributions

**Conceptualization:** Jichao Zhao, Vadim V. Fedorov.

**Data curation:** Roshan Sharma.

**Formal analysis:** Jichao Zhao, Roshan Sharma, Alexander Panfilov, Vadim V. Fedorov.

**Funding acquisition:** Jichao Zhao, Vadim V. Fedorov.

**Investigation:** Jichao Zhao, Roshan Sharma, Vadim V. Fedorov.

**Methodology:** Jichao Zhao, Roshan Sharma.

**Project administration:** Jichao Zhao, Vadim V. Fedorov.

**Resources:** Jichao Zhao.

**Software:** Roshan Sharma.

**Supervision:** Jichao Zhao.

**Validation:** Roshan Sharma, Anuradha Kalyanasundaram, Jieyun Bai, Ning Li, Alexander Panfilov.

**Visualization:** Roshan Sharma, James Kennelly.

**Writing – original draft:** Jichao Zhao.

**Writing – review & editing:** Jichao Zhao, Roshan Sharma, Anuradha Kalyanasundaram, James Kennelly, Jieyun Bai, Ning Li, Alexander Panfilov, Vadim V. Fedorov.

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
