## [Decision Letter · Decision Letter 0]

4 Jan 2023

Dear Dr. Zhao,

Thank you very much for submitting your manuscript "Mechanistic insight into the functional role of human sinoatrial node conduction pathways and pacemaker compartments heterogeneity: A computer model analysis" for consideration at PLOS Computational Biology.

As with all papers reviewed by the journal, your manuscript was reviewed by members of the editorial board and by several independent reviewers. In light of the reviews (below this email), we would like to invite the resubmission of a significantly-revised version that takes into account the reviewers' comments.

The paper is of interest, but in its current form the level of detail provided on model creation and validation are insufficient, so the quality of the paper can not be evaluated.

Please make sure that the data sources used to create and validate the model are referenced and that the species and temperature of any data sources are clearly stated.

Please also make sure that model code is available. Ideally submitted to a repository (cellml for example).

We cannot make any decision about publication until we have seen the revised manuscript and your response to the reviewers' comments. Your revised manuscript is also likely to be sent to reviewers for further evaluation.

Sincerely,

Steven A. Niederer

Guest Editor

PLOS Computational Biology

Daniel Beard

Section Editor

PLOS Computational Biology

The paper is of interest, but in its current form the level of detail provided on model creation and validation are insufficient, so the quality of the paper can not be evaluated.

Please make sure that the data sources used to create and validate the model are referenced and that the species and temperature of any data sources are clearly stated.

Please also make sure that model code is available. Ideally submitted to a repository (cellml for example).

Reviewer's Responses to Questions

**Comments to the Authors:**

Reviewer #1: The paper presents a detailed biophysical modelling study of pacemaking in the human sinoatrial node (SAN). The 2D SAN models are based on a mixture of experimental structure-functional data collected by the authors in previous studies and existing electrophysiology models and parameters from literature. Various pacemaking conditions are considered, including adenosine modulation, alterations of ionic currents, leading pacemaker shift, fibrotic remodelling, electrical insulation of the SAN and heaty failure. Strengths of the paper include the high level of detail in the model, it’s good agreement with a range of experimental observations and mechanistic explanations of some aspects of the human pacemaking. Weaknesses include a lack of clarity in explaining the underlying methods, in systemising/generalising the numerous modelling results and in delineating the novelty and importance of specific conclusions.

METHODS

“Ex-vivo Human SAN and Optical Mapping” and “Histological Imaging and Reconstruction of Human SAN” sections - please clarify what has been done in your previous studies, and what was done specifically for the current study; also clarify what types of data used in the models were ‘heart-specific’ and what types were generic (e.g., taken from previous experimental or modelling studies).

“The optically mapped 3D SAN activation maps were used to guide SAN histological dissection (Figure 1A)” – from Fig. 1 is it absolutely unclear how 3D activation maps were used.

“Subsequently, a semi-manual segmentation was performed on the stacks of Masson's trichrome” – please explain what is meant by ‘semi-manual’, e.g. what steps were not manual and how exactly they were performed.

“The resultant SAN computer model was obtained by projecting the 3D SAN model to the XY plane” – it’s unclear what is meant by ‘projecting’, why you didn’t you simply use the 2D slices seen in Fig. 1B?

“The insulating wall (at a uniform thickness of two pixels) between the SAN and RA was given a constant potential of -62.5 mV” – where did this value was taken from?

“…we considered the regional differences in gap junctional coupling between the SAN center, SAN head/tail, SACPs and RA tissues by setting the diffusion coefficients at a ratio of 2:3:6:14 in these regions. In the models, an anisotropic diffusivity ratio of 1:10 was used” – what were the actual (absolute) values of the diffusion coefficients?

RESULTS

Figures 2 and 3 – right panel in Fig. 2A and right panel in Fig. 3A both appear to show the same simulation (Ado 85%). However, in Fig. 2 this panel is match up with Ado 10um experimental data (where the leading pacemaker is shifted to the tail), whereas in Fig. 3 it is match up with Ado 100 um experimental data (where the leading pacemaker is shifted to the head most of the time) – with different conclusions drawn from these comparisons to the same Ado 85% simulation. Please used Ado 85% simulations consistently and make consistent conclusions.

One puzzling observation in Fig. 3 – in Ado 100 um experiment the leading pacemaker is shifted to the SAN head most of the time, and it makes sense that the earliest atrial activation is near there in the head; however, in the matching Ado 85% simulation the leading pacemaker is shifted to the tail, but the earliest activation site is still in the head, through the furthermost SACP. Any explanation?

In the light of my comments above, please rephrase the sentence “Somewhat similar trends were observed in the optically mapped human hearts ex-vivo at baseline and in the presence of low (10 µM) and high (100 µM) concentrations of adenosine (Figure 3B).”

Figure 4 – quite a few heterogeneity parameters are being varied, it could be better to organize numerical data into a summary figure (similar to how it’s done in Fig. 7) or a table, rather than write multiple numbers on top the figures; to a lesser extent, this might also be beneficial for Figs. 5-7.

Figure 7 – it’s unclear whether fibrotic remodelling was applied (and then reversed) in the surrounding atrial myocardium, or the SACPs only? In the latter case, why it wasn’t applied in the entire atrial tissue?

Figure 8, “In the computer model of the human SAN complex, a train of stimuli at a pacing cycle length of 500 ms was delivered from the right superior RA” – why this specific pacing site and this specific rate were chosen? Would the results be different if a different pacing setting is chosen? It might be necessary to perform some kind of sensitivity analysis to draw general conclusions in this case study.

DISCISSION

“The heterogeneity in expression of adenosine A1 receptors (A1R) or the IKACh channels within the human SAN pacemaker compartments explains leading pacemaker and preferential SACP shifts in the presence of adenosine” – this conclusion is very general, can it be broken down into more specific conclusions? In regard to the leading pacemaker, please also see my first set of comments on Results.

“The electrical insulation boundary between the SAN and RA except the SACP is required for normal SAN pacemaker and conduction function and to reproduce the leading pacemaker and the earliest atrial activation sites, observed in experimental and clinical studies” – this is one of the most important conclusions, please cite relevant experimental and clinical studies, either here or in Results.

“The INa current density and fibrotic remodelling (e.g. in heart failure HF) in SACPs modulate the SAN conduction (e.g. exit block) and the preferential SACP/exits to the atria (e.g. earliest atrial activation)” – this conclusion is interesting in the SAN context, but not at all surprising, since both blocking INa and increasing fibrosis are known to slow down or halt conduction. It’s worth at least referring to other (mechanistic) studies that explain this, e.g. for the SAN or/and for the atria.

“Human SAN pacemaker-conduction complex is vulnerable to reentrant arrhythmia and exit block under the mild intranodal INa current suppression and low-dose adenosine or HF-induced functional and structural remodeling” – again, this is quite general and may be worth breaking down into more specific conclusions. Besides, even without HF the SAN is prone to reentrant arrhythmia (Figure 8B), so it’s beneficial to explain how exactly HF makes the situation even worse (in a clinical sense).

“Critical Insights Learned and Clinical Implications” section – almost this entire section talks about pacemaking mechanisms, rather than clinical implications. Only the very last two sentences have some clinical relevance, but even they provide no specific clinical insights. I suggest to rename this section.

Reviewer #2: This paper presents a computer model of the sinoatrial node based on

histological data. Behaviour is compared with biological experiments. Based on

the model, several conclusions are made, including the effects of heart

failure, and SAN insulation. The SAN preparation and optical mapping are truly

at the forefront of the field. The model proposed is quite complex, and could

be of great potential use, but some of the assumptions need better

justification, putting the conclusions in doubt. Sensitivity analyses should be

performed to gauge the importance of parameters which have been estimated. The

paper is well written and argued, otherwise. Detailed comments follow.

The center of the SAN has a higher intrinsic frequency that the head and the

tail, which are treated as being the same. Do the authors have data to support

this? Traditionally, a monotonically decreasing intrinsic frequency gradient

has been described. The recent study of Brennan et al (JACC Clin Electrophys

2020) supports this notion. How do the ionic distributions used compare to

those in the aforementioned paper?

The authors do not mention adrenergic effects which counter sympathetic effects

but not always directly.

The authors modelled the insulative layer as a fixed voltage of -62.5 mV. Why

not just remove the layer as there will be coupling effects which should not be

present if truly isolated.

In Methods Realistic Human SAN Computer Model, what do the authors mean by

saying, "the neighboring RA and was much more efficient to run than a computer

model of the SAN directly based on the 3D histological data."

How do the authors justify the quantitative perturbations to the model under

HF?

There is confusion with regard to the A1R receptor density differences within

the SAN. The authors use a ratio of 10 between the center and the head/tail.

However, looking at the citation provided (Fig 6c), the ratio is well less than

2.

Reviewer #3: This looks interesting but cannot be reviewed in its current state:

1. The paper makes multiple claims to "present the first comprehensive biophysical computer model of the human SAN

complex based on direct molecular, structural and functional studies in the ex-vivo human heart", but very few details of the modelling are presented (no equations, no information on how equations were chosen or parametrised, no goodness of fit, no validation through novel predictions).

2. In addition to discussing such matters in the paper, the manuscript should provide code and data so that it complies with PLOS comp biol's data and code sharing policies.

**Have the authors made all data and (if applicable) computational code underlying the findings in their manuscript fully available?**

Reviewer #1: Yes

Reviewer #2: **No: **No references to the code or data seem to be listed.

Reviewer #3: **No: **

PLOS authors have the option to publish the peer review history of their article (what does this mean?). If published, this will include your full peer review and any attached files.

Reviewer #1: **Yes: **Oleg Aslanidi

Reviewer #2: No

Reviewer #3: **Yes: **Michael Clerx
---

## [Decision Letter · Decision Letter 1]

25 Apr 2023

Dear Dr. Zhao,

Thank you very much for submitting your manuscript "Mechanistic insight into the functional role of human sinoatrial node conduction pathways and pacemaker compartments heterogeneity: A computer model analysis" for consideration at PLOS Computational Biology.

As with all papers reviewed by the journal, your manuscript was reviewed by members of the editorial board and by several independent reviewers. In light of the reviews (below this email), we would like to invite the resubmission of a significantly-revised version that takes into account the reviewers' comments.

We cannot make any decision about publication until we have seen the revised manuscript and your response to the reviewers' comments. Your revised manuscript is also likely to be sent to reviewers for further evaluation.

Sincerely,

Steven A. Niederer

Guest Editor

PLOS Computational Biology

Daniel Beard

Section Editor

PLOS Computational Biology

Reviewer's Responses to Questions

**Comments to the Authors:**

Reviewer #1: The authors did a great job responding to my comments - I congratulate them on their interesting and substantial work.

My only remaining suggestion is to replace references 28 and 29 with more up-to-date publications by the same authors:

28. Morgan R, Colman MA, Chubb H, Seemann G, Aslanidi OV. Slow conduction in the border zones of patchy fibrosis stabilizes the drivers for atrial fibrillation: Insights from multi-scale human atrial modeling. Frontiers in Physiology 2016; 7: 474.

29. Roy A, Varela M, Chubb H, MacLeod RS, Hancox JC, Schaeffter T, Aslanidi O. Identifying locations of re-entrant drivers from patient-specific distribution of fibrosis in the left atrium. PLoS Computational Biology 2020; 16: e1008086.

Reviewer #2: The authors still need to be adequatelty respond to a couple of questions:

1) The intrinsic frequency gradient as assigned is not the traditional one and at odds with the recently published work of Brennan et al. The authors have not measured intrinsic frequency in any of the publications given as support. This point needs to be better discussed. Is this a prediction of the model that is unique for human SAN?

2) The space between the SAN and RA was not fully insulated but set as passive with a resting level of -62 mV. What biophysical mechanisms are in place to support this? It cannot just be assumed that since they are beside each other they will interact. Is the fat coupled electrically to the SAN and RA? What experimental proof is there for this coupling? Is this a prediction of your model? Why is this so important to the functioning of the model?

Reviewer #3: # Review 2

Thank you for providing more details of the used model(s).

The paper is a lot clearer on the used methods now, although two major points remain to be addressed in the methods section.

1. The overal level of detail in the methods section is still not sufficient. The study should be repeatable using the information provided in this section. In addition, the very interesting results and discussion show that important work has been done, so please describe this in greater detail!

2. A particular aspect is the "projection" or "shadow" of the 3d model. This is an interesting thing to do but it is not obvious to me that there is a single, obvious way to go about it. (I see it confused R1 too). This seems to be quite a vital part of your methodology ("which is crucial for the aims of this study") so please explain the steps you took in detail, with figures and/or supplementary figures if needed.

## Detailed points

### Introduction

> which consists of ~35-50% dense connective tissue.

Could you give some hint what the rest is composed of?

> This structure is ... necessary to maintain pacemaking and conduction

Please clarify whether you mean that this hetergogeneity is known to be present, or already know to be necessary.

> They established a foundation of theories

Who?

> developing new therapeutic approaches, e.g., biological pacemakers

Do you mean artificial? Either way, please provide one or more references for clarification.

> control simulation studies with only a varied certain contributing factor

What do you mean by a "varied certain contributing factor"? Please rewrite.

### Histological imaging and reconstruction of human SAN

> at a spatial resolution of 0.5*0.5um^2 using a 20 digital slide scanner

Is there a word missing after 20?

> High-resolutio fiber fields ... structure tensor.

Please explain in more detail or cite a reference.

### Realistic human SAN computer model

In what sense is the model "realistic"? I would omit this word as it seems like quite a subjective statement to me.

Explain _how_ a "shadow" was made. Is all information from all planes projected onto the same plane? If not, how do you choose which bits to keep? How do you deal with pixels that differed only in their Z coordinate?

Does the text describe all modifications that were made? Or did you make any further changes, e.g. external concentrations set to experimental values, cell capacitances adjusted for source/sink issues, temperature changed etc? Please rewrite the text so that it's a really clear list of exactly the steps that someone would need to get the same results as you.

Can you give some justification for the choice of models? I know this is a hard question and maybe quite subjective, but would be good to have a line or two on how you made this choice.

Thank you for adding code! Can you give a bit more detail of how you got from the CellML files to your C code? Was an automated tool used? Did you find any issues that needed fixing?

Did you pre-pace the system, i.e. run several beats before the ones you show in the simulations. If so, for how long? Does the system reach a "steady state" (mathematically a limit cycle) if you leave it running for e.g. 15 minutes of simulated time? How many beats were simulated before the results you show in each figure?

> It had relatively realistic geometrical loading

This is too vague and subjective: please replace this with something more precise and objective.

> We did not incorporate the SAN's internal blood vessels...focus of this study.

This explains why you omitted it, but not why it was OK to do so. I'm not asking you to add this in (the whole point of a model is to simplify!) but please add something (here or refer forward to the discussion) where you give us some idea of the impact of this choice. Especially given the source/sink issues, the size of the SAN, and the crucial isolation of the SAN from the RA it seems like this is not as easily ignored as in studies of e.g. the ventricles.

> and was given a constant potential between the...

Please add the exact potential, but also justify how you chose this.

I'm also interested in why you used a "neglectable" (I think the more common word here is "negligible"?) conductance instead of a zero conductance.

If, as you say, "this preset potential ... ensured a slight depolarization and hyperpolarization of neighboring RA and SAN cells" then by definition it wasn't negligible. Please explain what diffusivity you chose and how this affected the results.

> Modifications were made by...

Please list all modifications in the supplement.

If only the ratios of conductances were adjusted, then please adapt the text to make that clear.

Like one of the other reviewers mentioned, absolute values seem more useful than ratios here.

> The SACP cell models were not able to pace themselves

Was this by design or a result of your changes? Please make a clearer distinction between (1) what you did to make your model and (2) what your model then predicted.

> To incorporate the effects of adenosine/ACh into... concentration

I'm not sure what you mean by this. Please can you rewrite or add a one or two sentence explaination ("In brief, we...")

> changing the density of the IKACh current as a primary effector of...

Please rewrite to clarify this. I don't think you meant to say you "change it as a primary effector"?

> using the same formula by Grandi et al.

Please give this equation here or in the supplement (in which case you should refer to it here).

### SAN activation at baseline and...

> The developed ... model reliably reproduced

Please can you define this a bit more clearly? E.g. give real numbers etc. first before you summarise the overal performance this way; or end with a ":" and then list the reasons why you think it was reliable.

When you say "reproduced", do you mean that you made changes based on data (training), and then observed the model correctly predicting the higher-level behaviour (validation); or do you mean that you were able to tweak the model until it did (i.e. you calibrated it to the output data)? Either way this is a great achievement, but please specify it more clearly.

> The activation time... was about 75ms

Why "about"? What was it exactly?

> Qualitatively similar results

Please give exact numbers and allow the reader to judge this for themselves as well.

> which is slightly longer than in the model

Please give both numbers (experiment and model) in the text.

### The heterogeneity in expression of...

> To understand the effect of this heterogeneity on SAN function, we implemented...

Please give details in methods section and refer back to that here.

> we performed simulation in which the expression of A1R was...

Please add a line somewhere in the methods to say you treated expression as directly correlated with max conductance / permeability levels. This is quite uncontroversial, but still worth pointing out to e.g. experimental readers that the model does not actually include expression, translation, anchoring etc.

### The characteristics of SACPs dictate...

> Both INa and If currents ... were positively (or inversely) associated with the heart (or the SCL)

Had to read this twice. Maybe just write, "and inversely associated with the SCL"?

### The role of the insulation boundary...

> We found that computer models ... produced realistic activation time.

Please give exact numbers (model prediction and experimental equivalent or equivalents) so that readers can see if they agree that this was "realistic".

> a high chance of cardiac arrest

What do you mean in this case by "a high chance"? If I understand correctly it's a deterministic model, and you didn't re-run with different parameter settings etc?

Similarly on the next page "...increased the chance of SAN exit block", and later "had a higher tendency of exit block".

I would also advise to replace the term "cardiac arrest" here and in other places with "SAN arrest" as used later in the paper (there is no heart and no beating in this simulation).

### The impact of HF-induced...

> under HF remodelling (not shown).

If it's easy to do, please add this to the supplement.

### SAN is prone to...

> a train of stimuli

Please give amplitude and duration in supplement.

> in which we varied the degree of

Please add range over which it was varied.

> We found that depending on the parameters

Which ones and what values?

### Structural and electrical...

> However, no direct experimental data from larger species support this hypothesis.

Please clarify: no data exists at all, or existing data does not support?

### The novelty of...

> However, much less literature exists for human SAN

Could cite Noble et al. 2012 (https://doi.org/10.1113/jphysiol.2011.224238) here, and/or the Fabbri paper (which gives a nice review in the introduction).

> To date, there is no... that has integrated all anatomically...

Is "all" correct here? I'm sure it doesn't integrate all data?

### Critical functional insights...

> made possible by the higher A1R expression

higher IKAch conductance?

### Study limitations

> However, we do not think it...

This is not a good argument: It can still be relevant even if it's not the focus of your study.

### Conclusion

> Our novel... provides for the first time the crucial role

Other word instead of "provides"?

**Have the authors made all data and (if applicable) computational code underlying the findings in their manuscript fully available?**

Reviewer #1: Yes

Reviewer #2: Yes

Reviewer #3: Yes

PLOS authors have the option to publish the peer review history of their article (what does this mean?). If published, this will include your full peer review and any attached files.

Reviewer #1: **Yes: **Oleg Aslanidi

Reviewer #2: No

Reviewer #3: **Yes: **Michael Clerx
---

## [Decision Letter · Decision Letter 2]

24 Aug 2023

Dear Dr. Zhao,

Thank you very much for submitting your manuscript "Mechanistic insight into the functional role of human sinoatrial node conduction pathways and pacemaker compartments heterogeneity: A computer model analysis" for consideration at PLOS Computational Biology. As with all papers reviewed by the journal, your manuscript was reviewed by members of the editorial board and by several independent reviewers. The reviewers appreciated the attention to an important topic. Based on the reviews, we are likely to accept this manuscript for publication, providing that you modify the manuscript according to the review recommendations.

Sincerely,

Steven A. Niederer

Guest Editor

PLOS Computational Biology

Daniel Beard

Section Editor

PLOS Computational Biology

Reviewer's Responses to Questions

**Comments to the Authors:**

Reviewer #2: The authors should comment on the inferior-superior frequency gradient which they are ignoring. Their head and tail use the same cell model. The gradient has been shown to exist in humans by Brennan et al. and has important implications for the LPS.

Reviewer #3: Yes similar models have been around for some time, and yes we can read about previous work in previous papers, but this does not help the reader figure out what is done in *this paper* in any level of detail. It is remarkable to me that authors from the ABI, which spearheads many efforts to combat the reproducibility crisis in computational biology and cardiac electrophysiology in particular, would make such remarks.

**Have the authors made all data and (if applicable) computational code underlying the findings in their manuscript fully available?**

Reviewer #2: Yes

Reviewer #3: Yes

PLOS authors have the option to publish the peer review history of their article (what does this mean?). If published, this will include your full peer review and any attached files.

Reviewer #2: No

Reviewer #3: **Yes: **Michael Clerx

Figure Files:

Data Requirements:

Reproducibility:

References:

---

## [Editor Report · Decision Letter 3]

23 Nov 2023

Dear Dr. Zhao,

We are pleased to inform you that your manuscript 'Mechanistic insight into the functional role of human sinoatrial node conduction pathways and pacemaker compartments heterogeneity: A computer model analysis' has been provisionally accepted for publication in PLOS Computational Biology.

Best regards,

Steven A. Niederer

Guest Editor

PLOS Computational Biology

Daniel Beard

Section Editor

PLOS Computational Biology

---

## [Editor Report · Acceptance letter]

14 Dec 2023

PCOMPBIOL-D-22-01714R3 

Mechanistic insight into the functional role of human sinoatrial node conduction pathways and pacemaker compartments heterogeneity: A computer model analysis

Dear Dr Zhao,

I am pleased to inform you that your manuscript has been formally accepted for publication in PLOS Computational Biology. Your manuscript is now with our production department and you will be notified of the publication date in due course.

With kind regards,

Lilla Horvath
